# Spatial Zoning of Dry-Hot Wind Disasters in Shandong Province

**Nan Wang** [1], **Xiaoping Xue** [2,3,*], **Lijuan Zhang** [1,*], **Yue Chu** [1], **Meiyi Jiang** [1], **Yumeng Wang** [1], **Yiping Yang** [1], **Xihui Guo** [1], **Yufeng Zhao** [1] and **Enbo Zhao** [1]

[1] Heilongjiang Province Key Laboratory of Geographical Environment Monitoring and Spatial Information Service in Cold Regions, Harbin Normal University, Harbin 150025, China; wangnan199710@163.com (N.W.); cy254654@163.com (Y.C.); jiangmi199608@163.com (M.J.); wangyumeng19980603@163.com (Y.W.); ep_yangyiping@163.com (Y.Y.); 18235889220@163.com (X.G.); janekabesilsen@163.com (Y.Z.); zeb030410@163.com (E.Z.)

[2] Key Laboratory for Meteorological Disaster Prevention and Mitigation of Shandong, Jinan 250000, China

[3] Shandong Provincial Climate Center, Jinan 250000, China

[*] Correspondence: xxpdhy@163.com (X.X.); zlj@hrbnu.edu.cn (L.Z.)

**Abstract:** As a major agricultural province of China, Shandong province has long ranked first in agricultural growth value among all of the provinces; at the same time, it is also the province that is most affected by dry-hot wind. Therefore, it is of great significance to study the spatial zoning of the risks of dry-hot wind in this province. Based on meteorological, slope, and altitude data, and the principle of disaster risk assessment, this study uses a weighted comprehensive evaluation method, analytic hierarchy process, and ARC-GIS spatial analysis to study the spatial zoning of the risks of dry-hot wind in Shandong province. The results show that the high-risk regions of dry-hot wind are concentrated in the north-central portion of the province, the medium-risk regions are in the peripheral areas, and the low-risk regions are located mainly in the west, southwest, and east. Exposure of disaster-bearing bodies is high in the south and low in the north, while vulnerability to disaster-bearing bodies is high in the west and low in the east. The more developed areas in the east show high disaster prevention and mitigation capability, whereas this is weak in the west. In summary, dry-hot wind risk in Shandong province varies significantly by area. The medium- and high-risk areas are mainly in the west and central portions of the province.

**Keywords:** dry-hot wind disaster; risk zoning; Shandong province; natural disaster risk assessment principle

## 1. Introduction

Dry-hot wind is a type of severe agricultural wind disaster with high temperature and low humidity. The late spring and early summer are the seasons when the direct sunlight in the northern hemisphere is the greatest, and the weather is sunny and drier before the arrival of the northern rainy season. Under the control of the dry air mass, the sky is fine, dry, and windy, and there are few opportunities for cloud formation to cause rain, so it is easy to form dry-hot winds. It causes low humidity in the air and water in the soil to evaporate, severely impacting crop yields and economic development [1,2]. The dry-hot wind disaster has become an important factor restricting the growth and development of crops by intensifying plant transpiration, resulting in insufficient water supply to the roots, causing an imbalance of water and nutrients in the plant. Typically, the leaves change color and normal physiological activities of the plant are damaged or inhibited, resulting in a significant shortening of the filling period of the crop, the high temperature brought by the dry hot wind after flowering will shorten the growing period of the seeds, and damage to protein and starch structure [3–8]. When the risk of dry-hot wind is low, wheat yield can decline by 5–10%, and in severe cases, by 20–30% or more [9].

In the 1950s, the Soviet Union conducted preliminary research on dry-hot wind disasters, which was at the leading level compared with other countries, mainly on the formation indicators and causes of dry-hot wind, spatial distribution, and that of disaster prevention and mitigation measures [10]. In recent years, more scholars have paid attention to the occurrence regularity of the number and intensity of dry-hot winds. Tavakol et al. [11] analyzed the spatial patterns and temporal changes of hot, dry, and windy events (HDWs) in the central United States for two time periods: 1949 to 2018 (70 years) and 1969 to 2018 (50 years). рокопец et al. [12] evaluated the dynamic variation characteristics of the total number and intensity of dry-hot winds over the lower Volga River. In contrast, Chinese scholars have conducted relatively more studies on the occurrence of hot-dry wind. Hou et al. [13] considered that the occurrence characteristics of dry-hot wind in the Hexi region of Gansu province and surrounding areas during June and July from 1960 to 2017 showed a tendency to decrease slowly and then increase rapidly. Cheng et al. [14] pointed out the occurrence frequency of different graded of the hazard in Henan province is on the rise. You et al. [15] analyzed the temporal-spatial distribution of dry-hot wind in the Hebei province winter wheat region during the past 35 years. The Huang-Huai-Hai region is the region with the most frequent occurrence of hot-dry winds in China, so it has attracted the attention of scholars. Li et al. [16] considered the annual average number of regional DHW events in the Huang-Huai-Hai Region showed a decreasing trend from 1961 to 2010 and increased in 2011–2018. Shi et al. [17] also pointed out the annual average of light and serve dry-hot wind in the Huang-Huai-Hai plain declined from 1963 to 2012. Zhao et al. [18] studied the spatial-temporal changes of dry-hot wind of winter wheat in the Huang-Huai-Hai plain under climate change. Wheat is the crop most affected by hot and dry wind, so scholars have focused on the analysis of the impact of hot and dry wind on wheat. Chen et al. [19] analyzed the influence of dry-hot wind on the wheat in Henan province and proposed that the number of dry-hot wind days has a significantly negative correlation with wheat meteorological yield. Shi et al. [20] proposed that the total number of dry-hot wind days had a negative correlation with thousand grain weight of winter wheat in Hebei province. Yang et al. [21] constructed a dry-hot wind risk assessment index system, including a dry-hot wind intensity risk index and a comprehensive disaster resistance index by using the meteorological data, yield and structure data, and development period data of the winter wheat observation station, and established a dry-hot wind risk assessment model. The risk of the dry-hot wind in the main winter wheat producing areas in North China was assessed, and the results showed that southeastern Hebei and northwestern Shandong were high-risk areas, while southern Henan, eastern Shandong, and eastern Hebei were low-risk areas. Based on the theory of agrometeorological disaster risk analysis, Chen et al. [22] analyzed the influence degree and risk probability of dry-hot wind on wheat yield in wheat production in Henan province by constructing a disaster function and using EOF and probability analysis methods. The results showed that dry-hot wind was the main disaster that influenced the high and stable yield of wheat in most of Henan province. In order to further study the resistance to dry-hot wind of wheat, Juraev et al. [23] planted varieties and lines in November in the late sowing period. The daily temperature, wind speed, and relative humidity were selected to compare and study the changes of plant height, ear length, and grain number per ear of wheat varieties in Casdalia and Surcandalia during their developmental stages. The results showed that dry-hot wind had a significant effect on the traits of wheat varieties and lines. Wang et al. [24] proposed a framework (DID) to quantify the impact of dry-hot wind on winter wheat in northern China and the framework can effectively detect winter wheat growing areas affected by dry-hot wind hazards. The estimated damage showed a notable relationship ($R^2 = 0.903$, $p < 0.001$) with the dry-hot wind intensity calculated from meteorological data. Deng et al. [25] comprehensively summarized the causes, protecting technology and answering tactics of dry-hot wind disasters.

With the deepening of research, scholars' research on hot-dry wind has shifted from occurrence to a disaster defense system, and disaster risk zoning is the basis for establishing

a disaster prevention and mitigation system. Natural disaster zoning is the division of regions based on the temporal and spatial distribution of the occurrence and development of natural disasters; it can provide a scientific basis for regional disaster prevention and mitigation. In fact, strengthening the research on the comprehensive zoning of natural disasters is listed in China's 21st Century Agenda [26]. For example, Cheng et al. [27] established the index of yield loss risk of dry-hot wind and integrated to zone the comprehensive hazard risk in Henan province. Wu et al. [28] developed a new regionalization method, wherein type one is high temperature and low humidity and type two is immature death after rain, for the dry-hot windy days regionalization in the NCP. According to the risk assessment theory of natural disasters, some scholars have performed fine zoning and assessment of dry-hot wind risks in the winter wheat region of Henan province [29] and spring wheat region of Inner Mongolia [30] from four aspects: risk of disaster-causing factors, vulnerability of disaster-pregnant environment, exposure of disaster-bearing body, and ability of disaster prevention and mitigation. As a major agricultural province in China, Shandong province has a winter wheat planting area up to $4.003 \times 10^6$ hm$^2$ and an annual yield of $2.472 \times 10^7$ T [31]. Shandong province is located in the Huang-Huai-Hai region of China, which is the region most affected by hot-dry wind disasters. However, there are few studies on the risk of hot-dry wind in Shandong province. Therefore, it is urgent to carry out the research on the risk regionalization of hot-dry wind in Shandong province. Based on disaster risk theory, we analyze dry-hot wind disaster from four perspectives: risk, exposure, vulnerability, and disaster prevention and mitigation capability. A dry-hot wind disaster risk index model is established, and spatial zoning of dry-hot wind disasters in Shandong is examined using ARC-GIS spatial analysis, weighted comprehensive evaluation method, and analytic hierarchy process. The aims of this study are to comprehensively assess and zone the risk of dry-hot wind, so as to ensure the safety of wheat production. In order to reduce disaster risk and provide reference for agricultural production layout and scientific decision-making, dry-hot wind risk zoning is of great significance to regional agricultural management and production, people's lives, and food security.

## 2. Materials and Methodology

### 2.1. Study Area

Shandong province is located on the east coast of China and the lower reaches of the Yellow River (114°48′ E–122°42′ E and 34°23′ N–38°17′ N), as shown in Figure 1. The total land area is 157,900 km$^2$. The climate type is warm temperate monsoon. Precipitation is concentrated, and rain and heat occur in the same season. Spring and autumn are short, while winter and summer are long. The annual average temperature range is 11–14 °C and the annual average precipitation range is 550–950 mm. The rainfall season is unevenly distributed, with 60–70% of annual precipitation in summer. Landform types include plains, terraces, hills, and mountains. There is a dense river network in the region, including the Yellow River, Huaihe River, Haihe River, and smaller rivers in the central and southern mountainous area.

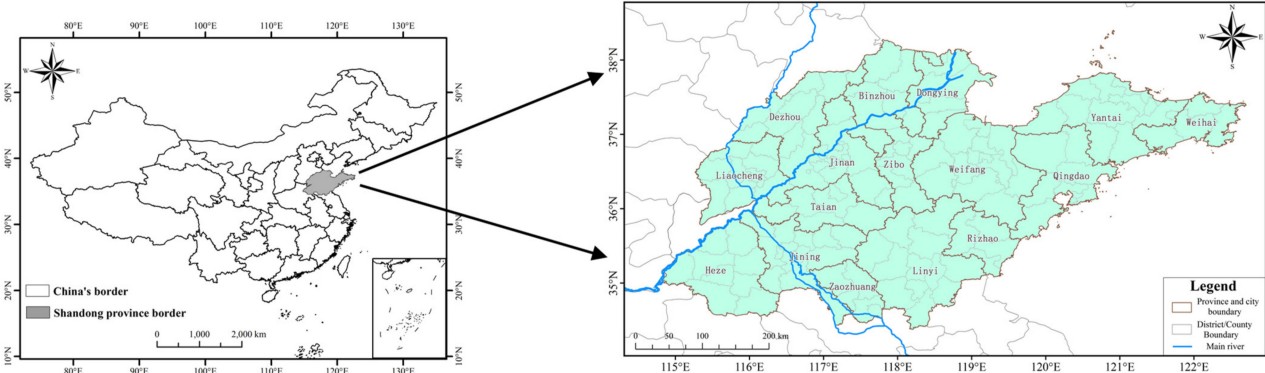

**Figure 1.** Administrative division of Shandong province, China.

## 2.2. Data Sets

The meteorological data used here include wind speed, temperature, and relative humidity from 1991 to 2020. The aspect, altitude, slope, river network density, and land-use type are obtained from the Meteorological Information Center of Shandong Meteorological Bureau. Total GDP, total population, administrative area, wheat planting area, crop planting ratio, population density, education level, per capita GDP, and crop planting area are from the 2018–2020 Statistical Yearbook of Shandong province.

## 2.3. Methods

Based on the natural disaster risk theory, this paper constructs a scientific framework as follows (Figure 2):

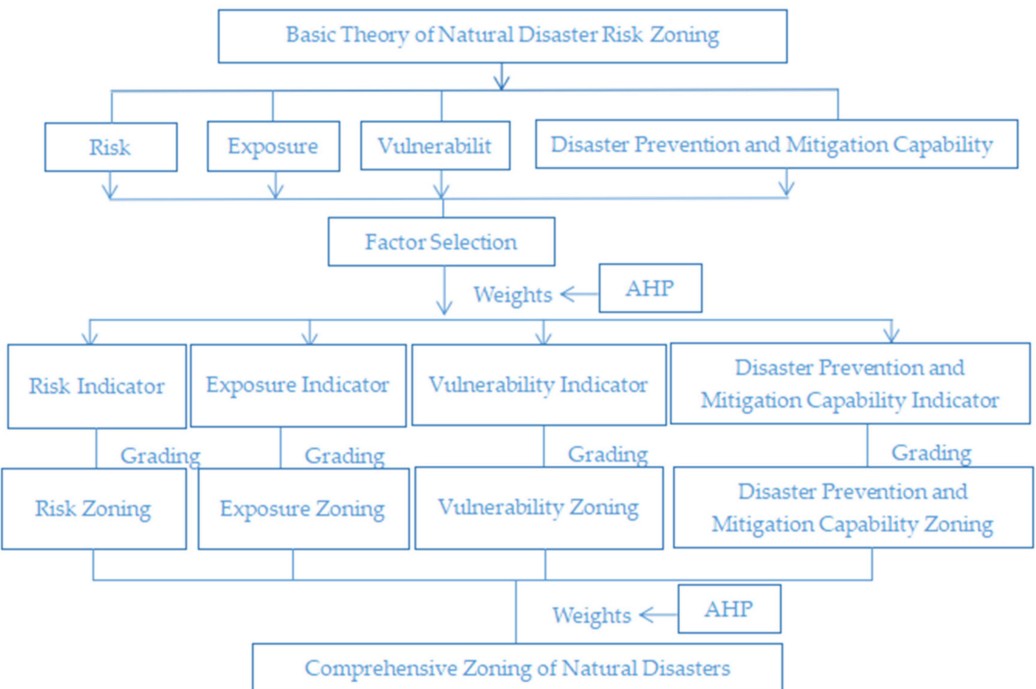

**Figure 2.** Scientific framework or spatial zoning of hot-dry wind disasters in Shandong province.

### 2.3.1. Basic Theory of Disaster Risk Assessment

Based on the theory of natural hazard risk formation [32], meteorological hazard risk is formed by the combination of four components: hazard (causative factor), exposure (carrier), vulnerability (carrier), and prevention and mitigation capacity. Each factor is in turn composed of a series of subfactors. The expressions are:

Disaster risk index = f (hazard, exposure, vulnerability, disaster prevention and mitigation capacity)    (1)

Hazardous factors: Hazardous factors include meteorological factors and environmental sensitivity. All meteorological factors that may lead to disasters can be called meteorological factor hazards; the sensitivity of the pregnant environment refers to the degree of strengthening or weakening of meteorological factors in the natural surface environment.

Exposure of disaster-bearing body: Disaster-bearing body is the object of disaster-causing factors and is the entity that bears the disaster. Exposure of the hazard-bearing body is the result of the interaction between the hazard-causing factor and the hazard-bearing body, and the exposure of the hazard-bearing individual to the hazard-causing factor.

Vulnerability of the disaster-bearing body: A disaster can be formed only when it acts on the corresponding object, i.e., human beings and their socioeconomic activities. Specifically, it refers to the degree of hazard or loss caused by the potential risk factors

for all objects that may be threatened by the disaster-causing factors that exist in a given hazard area, and its combination reflects the degree of loss from meteorological disasters.

Prevention and mitigation capacity: This refers to various management measures and countermeasures used to prevent and mitigate meteorological hazards, including management capacity, mitigation input, and resource preparation. The more proper management measures and strong management capacity, the less potential losses may be suffered and the less risk of meteorological disasters.

Based on the above theory, the hierarchical structure model of dry-heat wind risk assessment was constructed. The risk index values of hazard factor, exposure factor, vulnerability factor, and disaster prevention and mitigation capacity of dry-heat wind disaster are calculated as *Ya*, *Yb*, *Yc*, and *Yd*, respectively, by the weighted comprehensive evaluation method. Through the natural disaster risk index formula, combined with the dry-heat wind disaster assessment index system of Shandong province, its disaster risk index model is:

$$F = W_a \times Y_a + W_b \times Y_b + W_c \times Y_c + W_d \times Y_d \tag{2}$$

In which, *F* is the dry heat wind hazard risk index, which indicates the degree of dry heat wind hazard; the larger the value of *F*, the higher the risk and the opposite the lower. *y* is the risk index value of hazard factor, exposure factor, vulnerability factor, and prevention and mitigation capacity of dry heat wind hazard. *w* is the weight of each index.

Based on the composition of meteorological data, topography, and socioeconomic elements, Figure 3 below shows the hierarchical structure model of dry-heat wind disaster risk assessment in Shandong province.

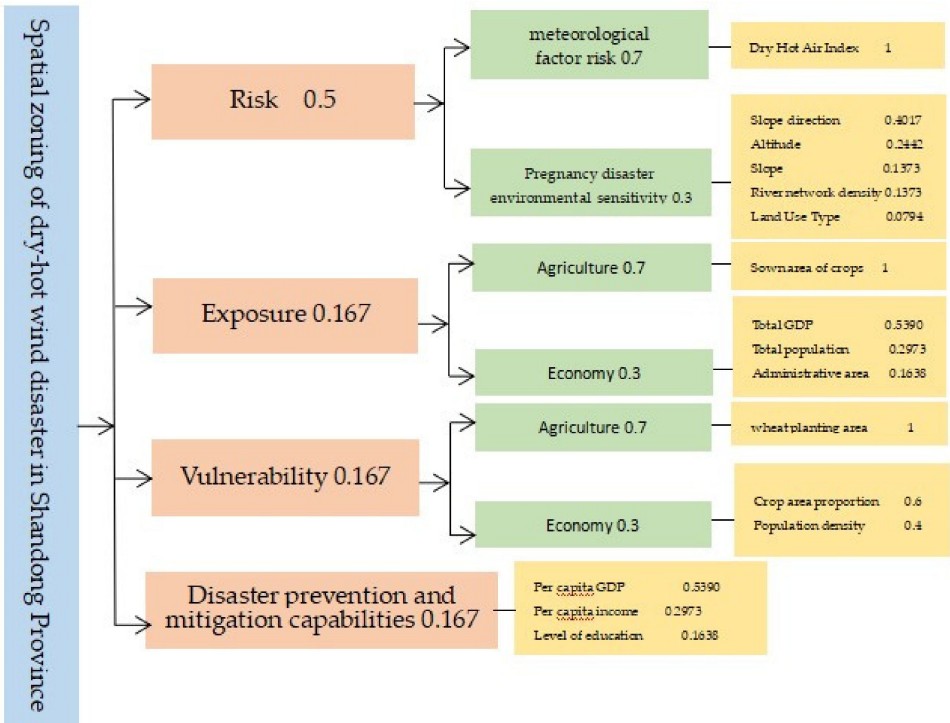

**Figure 3.** Hierarchical model of dry hot air risk assessment.

2.3.2. Weighted Comprehensive Evaluation Method

The weighted comprehensive evaluation method is a method that solves the "bottom-up" indexes in the risk hierarchy analysis and evaluation model, which considers the degree of influence of each factor on the overall object and integrates the strengths and weaknesses of each specific index and uses a numerical index to focus on the strengths and weaknesses of the entire evaluation object. This method is especially suitable for

comprehensive analysis and evaluation of technology, strategy, or programs and is one of the most commonly used calculation methods. Its expression is:

$$Y_i = \sum_{j=1}^{m} \lambda_j X_j \ \ i = 1, 2, \dots, m; \ j = 1, 2, \dots, m \tag{3}$$

where $Y_i$ denotes the disaster risk index, $i$ denotes hazard, exposure, vulnerability, and disaster prevention and mitigation capacity, respectively; $X_j$ is the factor affecting hazard, exposure, vulnerability, and disaster prevention and mitigation capacity, and $\lambda_i$ is the weight value ($0 \le \lambda_j \le 1$).

For the comprehensive risk index of natural disasters, the expressions are:

$$Y = \sum_{i=1}^{n} W_i Y_i \ \ \ i = 1 \tag{4}$$

where $Y$ denotes the comprehensive disaster risk index; $Y_i$ is the hazard index, exposure index, vulnerability index, and disaster prevention and mitigation capacity index, and $W_i$ is the weight value.

The stronger the disaster prevention and mitigation capacity is, the smaller the comprehensive risk index is, so the "negative sign" is used.

Where $\lambda_j$ and $W_i$ are determined using hierarchical analysis, as described in research Section 2.3.3, each factor in the formula needs to be standardized because of different dimensions; see research Section 2.3.4 for details.

### 2.3.3. Analytic Hierarchy Process

Analytic Hierarchy Process (AHP) is a simple method for making decisions on some more complex and vague problems, especially for those problems that are difficult to fully quantitatively analyze [33]. This paper uses the operation principle of the analytic hierarchy process and uses the 1–9 scale method given by Saaty to construct the judgment matrix for the pairwise relationship of the influence factors. The pairwise comparison of all influence factors determines the weight of each influence factor, which avoids the result error caused by the subjectivity of the expert. The qualitative comparison scale values between the two influencing factors are shown in Table 1 below:

**Table 1.** Scale of AHP analysis method.

| Scale $b_{ij}$ | Definition |
|:---:|:---:|
| 1 | The *i* factor is as important as the *j* factor. |
| 3 | The *i* factor is slightly more important than the *j* factor. |
| 5 | The *i* factor is more important than the *j* factor. |
| 7 | The *i* factor is much more important than the *j* factor. |
| 9 | The *i* factor is absolutely more important than the *j* factor. |
| 2, 4, 6, 8 | Between the noted levels. |

Solve the maximum eigenvector value of the judgment matrix and its corresponding eigenvector by the sum-product method and check the consistency of the matrix (the following formula): After passing, solve it by the sum-product method.

$$CI = \frac{\lambda_{\max} - n}{n-1} = \frac{-\sum_{i=1}^{n} \lambda_i}{n-1} \tag{5}$$

$$CR = \frac{CI}{RI} < 0.1 \tag{6}$$

In the formula, $CI$ is the consistency index of the judgment matrix, $\lambda_{\max}$ is the largest characteristic root of the matrix, $n$ is the order of the discrimination matrix, $CR$ is the random

consistency index of the judgment matrix, and *RI* is the average random consistency index of the discrimination matrix. The values of *RI* are shown in Table 2:

**Table 2.** Numerical values of random consistency index *RI*.

| M | 1 | 2 | 3 | 4 | 5 | 6 | 7 | 8 | 9 | 10 | 11 |
|---|---|---|---|---|---|---|---|---|---|----|----|
| *RI* | 0.00 | 0.00 | 0.58 | 0.90 | 1.12 | 1.24 | 1.32 | 1.41 | 1.45 | 1.49 | 1.51 |

This paper adopts the Analytic Hierarchy Process (AHP), taking the sensitivity of the disaster-pregnant environment as an example, and constructs the judgment matrix of each index and the calculation results are shown in Table 3. Since *CR* < 0.1, the matrix passed the consistency test.

**Table 3.** Judgment matrix and weights of various perceptual factors.

| | I | II | III | IV | V | Weights (W) | Matrix Product (AW) | AW/W | $\lambda_{max}$ | CI | CR |
|---|---|---|---|---|---|---|---|---|---|---|---|
| I | 1 | 2 | 3 | 3 | 4 | 0.402 | 2.03 | 5.05 | 5.033 | 0.008 | 0.007 |
| II | 1/2 | 1 | 2 | 2 | 3 | 0.244 | 1.23 | 5.04 | | | |
| III | 1/3 | 1/2 | 1 | 1 | 2 | 0.137 | 0.689 | 5.03 | $\lambda = \sum(AW/W)/n$ | | |
| IV | 1/3 | 1/2 | 1 | 1 | 2 | 0.137 | 0.689 | 5.03 | $CI = (\lambda - n)/n - 1$ | | |
| V | 1/4 | 1/3 | 1/2 | 1/2 | 1 | 0.079 | 0.399 | 5.05 | $RI = 1.12$ | | |
| | | | | | | | | 25.18 | $CR = CI/RI$ | | |

Note: In the table, I. Slope direction, II. Elevation, III. Slope, IV. River network density, V. Land-use type.

Similarly, the weights of each factor of hazard, exposure, vulnerability, disaster prevention and mitigation capacity, and combined disaster risk were obtained as shown in Figure 3.

### 2.3.4. Standardization

In the process of zoning, the different magnitudes of the selected factors lead to a large difference in order of magnitude; for example, the total population is 3,923,000 people, while the total GDP is about 302.22 billion yuan, so when calculating the hazard index of disaster-causing factors, normalization is required so that the values of each factor are between 0 and 1. Furthermore, when assessing the hazard of disaster-causing factors, exposure of disaster-bearing bodies, vulnerability of disaster-bearing bodies, and disaster prevention and mitigation capacity, the larger the number of influencing factors, the larger the hazard of disaster-causing factors, exposure of disaster-bearing bodies, vulnerability of disaster-bearing bodies, and disaster prevention and mitigation capacity, while some factors are the opposite. Therefore, in the assessment process, the criterion of a great value or the criterion of a very small value should be standardized first, and the formula is as follows. For example, the greater the dry and hot wind index, the greater the hazard of disaster-causing factors, so choose the great value standardization for the dry and hot wind index, and choose Equation (7); for example, the greater the slope, the less the sensitivity of disaster-preventing environment, so standardize the slope for the small value, and choose Equation (8).

Maximum standardization:

$$X'_{max} = \frac{\left|X_{ij} - X_{min}\right|}{X_{max} - X_{min}} \tag{7}$$

Minimum standardization:

$$X'_{min} = \frac{\left|X_{max} - X_{ij}\right|}{X_{max} - X_{min}} \tag{8}$$

where $X_{ij}$ is the index number of the *j*-th factor of the *x* factor; $X'_{max}$ and $X'_{min}$ are the dimensionality of $X_{ij}$; $X_{max}$ and $X_{min}$ are the minimum and maximum values in the index sequence.

### 2.3.5. Arc-GIS Spatial Analysis

The spatial analysis methods involved in this paper include Kriging interpolation, spatial reclassification, spatial raster calculation, and slope extraction. The raster resolution is 100 m × 100 m.

Kriging interpolation is a method of unbiased optimal estimation of regionalized variables in a limited region based on the variogram theory and structural analysis. Not only can it reflect the spatial structure characteristics of variables, but it can also reflect the random distribution characteristics of variables [34]. There are many factors that affect the spatial change of natural geographical elements. The comprehensive action of these factors forms the zonal regularity on the Earth's surface, and the natural geographical elements are also disturbed by various random factors. Geographers try to explore the regional regularity of their natural geographical elements and strive to minimize the interference of random factors. Kriging interpolation can minimize the interference of random factors with the help of the optimal method. Therefore, this method can be used to analyze the changes of elements in the interpolation space, such as the change of temperature, the regional distribution law of water quality, vegetation, soil, and other elements with zonal distribution law [35]. Based on the above research conclusions, this paper uses the Kriging interpolation method to interpolate the zonal geographical elements. According to these research results, this paper applies the Kriging interpolation method to zoning elements and zoning results.

In this paper, the natural breakpoint method is used for classification in risk, exposure, vulnerability, disaster prevention and mitigation capacity and comprehensive risk. In fact, there are many classification methods, such as equal division, standard deviation classification, and so on. Fu et al. [36] and others have concluded that the natural breakpoint method can adequately extract the useful information contained in the index, so as to establish a more reasonable and accurate index evaluation space. At the same time, the risk, exposure, vulnerability, disaster prevention, and reduction ability and comprehensive risk are divided into low, medium, and high levels, mainly because if there are too many levels, the spatial expression effect is not clear enough.

### 3. Results

#### 3.1. Spatial Distribution of Dry-Hot Wind Risk

The risk of dry-hot wind includes the risk of meteorological factors and the disaster environment sensitivity, and the risk of meteorological factors is the main factor constituting the risk of dry-hot wind. Disaster environment sensitivity can aggravate or reduce the risk of meteorological factors. At the same time, this paper also consulted relevant experts, such as Shandong Meteorological Bureau and the Department of Agriculture, and gave the risk of meteorological factors and the disaster environment sensitivity the weights of 0.7 and 0.3, respectively. The weight in the composition of dry-hot wind meteorological factor risk index is explained in the fourth question (Figure 3).

#### 3.1.1. Zoning of Meteorological Factor Risk

According to the ground meteorological observation specification of the People's Republic of China (Table 4), soil relative humidity at 20 cm, daily maximum temperature (°C) air relative humidity at 14:00 (%) and wind speed at 14:00 (m/s) were selected as the grade indicators of dry-hot wind. Since the meteorological station does not observe 20 cm soil relative humidity, there is no 20 cm soil relative humidity in the meteorological observation records. Considering that precipitation is the main factor affecting 20 cm soil relative humidity, the maximum process precipitation in early and middle May was used to replace 20 cm soil relative humidity in this study. The classification standard of dry-hot wind used in this study is shown in Table 5.

**Table 4.** High temperature and low humidity type dry-hot wind grade indicators.

| Area | Mild | | | | Medium | | | Severe | | |
|------|------|------|------|------|--------|------|------|--------|------|------|
| | 20 cm Soil Relative Humidity | Daily Maximum Temperature (°C) | Air Relative Humidity at 14:00 (%) | Wind Speed at 14:00 (m/s) | Daily Maximum Temperature (°C) | Air Relative Humidity at 14:00 (%) | Wind Speed at 14:00 (m/s) | Daily Maximum Temperature (°C) | Air Relative Humidity at 14:00 (%) | Wind Speed at 14:00 (m/s) |

**Table 5.** Dry-hot wind generation index.

| Maximum Process Precipitation in Early and Mid-May | The Time Period is Mid to Late May | | | | | | | | |
|---|---|---|---|---|---|---|---|---|---|
| | Mild | | | Medium | | | Severe | | |
| | Daily Maximum Temperature (°C) | Air Relative Humidity at 14:00 (%) | Wind Speed at 14:00 (m/s) | Daily Maximum Temperature (°C) | Air Relative Humidity at 14:00 (%) | Wind Speed at 14:00 (m/s) | Daily Maximum Temperature (°C) | Air Relative Humidity at 14:00 (%) | Wind Speed at 14:00 (m/s) |
| <25 mm | 31 | ≤30 | ≥2 | ≥32 | ≤25 | ≥3 | ≥35 | ≤25 | ≥3 |
| ≥25 mm | ≥33 | ≤30 | ≥3 | ≥35 | ≤25 | ≥3 | ≥36 | ≤25 | ≥3 |

The harmful degrees of the days of mild, medium, and severe dry-hot wind are different. The more instances of severe dry-hot wind that happen, the stronger the influence of dry-hot wind is in the area. Therefore, when constituting the dry-hot wind index, different weights must be given to the days of mild, medium, and severe dry-hot wind. According to the basic principle of AHP, when calculating the weight, first, the ratio matrix is constructed according to the scale grade table. Since the severe dry-hot wind is very important compared with the mild dry-hot wind, the ratio is assigned as one. The severe dry-hot wind is slightly more important than the medium dry-hot wind, so the ratio is assigned as two. The medium dry-hot wind is slightly more important than the mild dry-hot wind, so the ratio is assigned as three, so the ratio matrix is formed (as shown in the Table 6 below). The consistency test index *CR* of the matrix was calculated as *CR* = 0.08. Because *CR* < 0.1, the matrix passed the consistency test. Using the sum product method, the weights of the times of mild, medium, and severe dry-hot wind are 0.2, 0.3, and 0.5, respectively.

**Table 6.** Judgment matrix and weights of various perceptual factors.

| | I | II | III | Weights (W) | Matrix Product (AW) | AW/W | $\lambda_{max}$ | CI | CR |
|---|---|---|---|---|---|---|---|---|---|
| I | 1 | 2 | 3 | 0.5 | 1.62 | 3.01 | 3.01 | 0.004 | 0.008 |
| II | 1/2 | 1 | 2 | 0.3 | 0.89 | 3.00 | $\lambda = \sum(AW/W)/n$ | | |
| III | 1/3 | 1/2 | 1 | 0.2 | 0.49 | 2.99 | $CI = (\lambda - n)/n - 1$ | | |
| | | | | | | 9 | $RI = 0.58$ | | |
| | | | | | | | $CR = CI/RI$ | | |

Note: I. Severe dry-hot wind days, II. Moderate dry-hot wind days, III. Mild dry-hot wind days.

According to Table 5, the number of hot-dry days at all levels from 1991 to 2020 was calculated. In the risk zoning of dry-hot wind days, the harm degrees of mild, medium, and severe dry-hot wind days are different. The more severe dry-hot wind days, the stronger the impact of dry-hot wind. Therefore, the mild, medium, and severe dry-hot wind days constitute a comprehensive risk index, and different weights are assigned to the mild, medium, and severe dry-hot days, which are 0.2, 0.3, and 0.5, respectively.

$$R = 0.2D_1 + 0.3D_m + 0.5D_s \tag{9}$$

where $R$ is the comprehensive index of dry-hot wind (d), $D_1$ is average number of days of mild dry-hot wind disaster during the 30-year study period (d), $D_m$ is the average number of days of medium dry-hot wind disaster during the 30-year study period (d), and $D_s$ is the average number of days of severe dry-hot wind disaster during the 30-year study period (d).

To sum up, mild, medium, and severe dry-hot wind refers to the results obtained by combining different meteorological indicators. The comprehensive index is a linear

addition of the days of mild, medium, and severe dry-hot wind, which is formed for the risk zoning index.

The risk of dry-hot wind is composed of two parts: the risk of meteorological factors and the pregnancy disaster environmental sensitivity. Dry-hot wind is a type of meteorological disaster with high temperature, low humidity, and a certain wind force. The topographic factors in the pregnancy disaster environmental sensitivity affect the hazard degree of dry-hot wind. Since meteorological factors are more important compared with topographic factors, the risk of meteorological factors is given a higher weight. According to the scale of the AHP analysis method, when the *i* factor is more important than the *j* factor, the weight of the *i* factor is given to 0.7, and the weight of the *j* factor is given to 0.3. Therefore, the weight of the risk of meteorological factors is assigned 0.7, and the weight of the pregnancy disaster environmental sensitivity is assigned 0.3.

The spatial distribution of the average number of days of dry-hot wind in Shandong province over the past 30 years is shown in Figure 4. The spatial distribution of the days of mild dry-hot wind is similar to that of moderate dry-hot wind. High values are distributed mainly in the central area, and low values are found in the east and west. The highest values are 1.1 d and 0.4 d, and the lowest value is 0. Areas with a high number of days with severe dry-hot wind disasters are concentrated in the north, and the highest figure is 0.4 d. In the remaining areas, the number of days with severe dry-hot wind disaster is significantly reduced, with the lowest value at 0. The spatial distribution of the index varies significantly. The high-value areas are concentrated in Weifang, Zibo, Jinan, Binzhou, and Dongying. The highest number of days is 1.9, and the lowest number of days is 0.

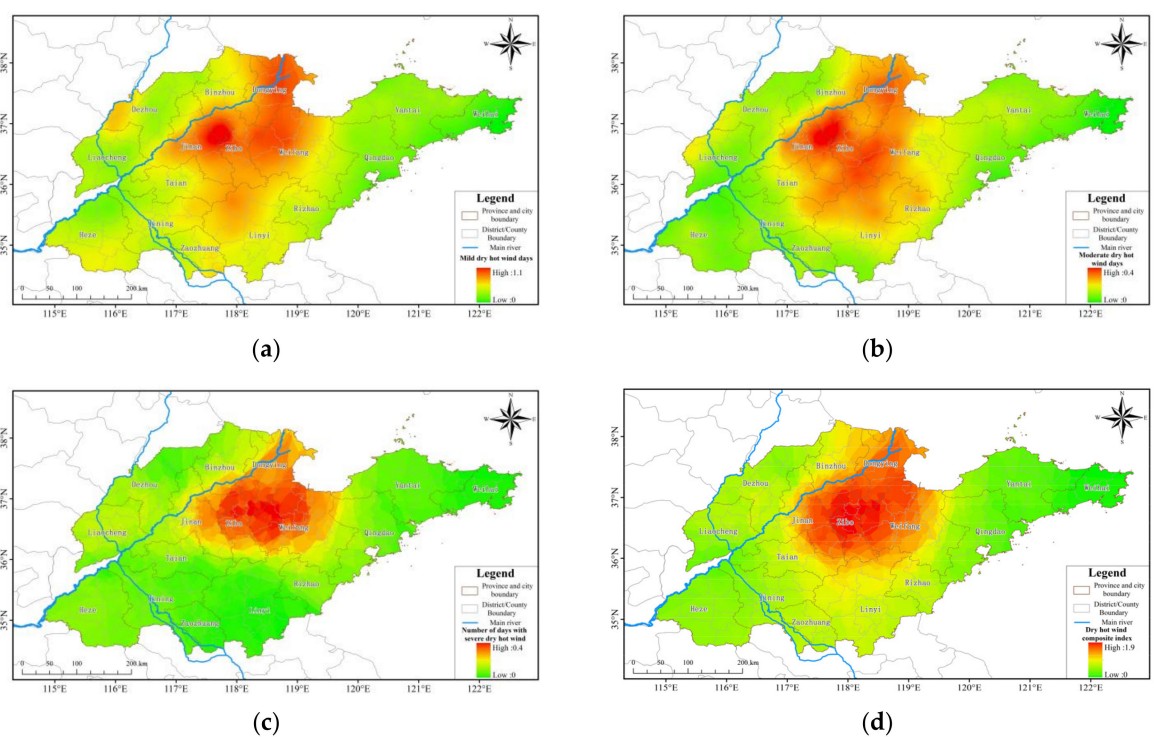

**Figure 4.** Spatial distribution of dry-hot wind meteorological factors in Shandong province: (**a**) mild dry-hot wind days; (**b**) moderate dry-hot wind days; (**c**) number of days with severe dry-hot wind; (**d**) dry-hot wind composite index.

### 3.1.2. Zoning of Disaster Environment Sensitivity

Aspect, altitude, slope, river network density, and land-use type are selected as the zoning indices for disaster environment sensitivity (Figure 5). The south slope has longer sunshine duration and higher temperature, so the closer to the south slope, the more dangerous the hot-dry wind will be. The southwest slope and southeast slope also receive more solar radiation relatively, so the sensitivity is also higher. The east slope warms

faster than the west slope, so the sensitivity is higher than that of the west slope; thus, that of the northeast slope is also slightly higher than that of the northwest slope. The temperature of the north slope is the lowest, so the sensitivity is the lowest. The sensitivity of slope-free area is slightly lower than that of the south slope and higher than that of the east slope. Therefore, the ordering and scoring of the slope aspect are shown in Table 7. Temperature decreases with increased altitude; therefore, the higher the altitude, the lower the temperature. The greater the slope is, the less solar radiation per unit an area receives, and therefore the lower the sensitivity would be [37]. The slope range in Shandong province is 0–49.5°, so every 10° is assigned a grade, as shown in Table 8. In addition, the higher the river network density, the higher the air humidity, and the less the impact from dry-hot wind. Compared with unused land, woodland and grassland have better water conserving capacity and higher air humidity, which helps to reduce the influence of dry-hot wind disasters. The scores for different land use types are shown in Table 9.

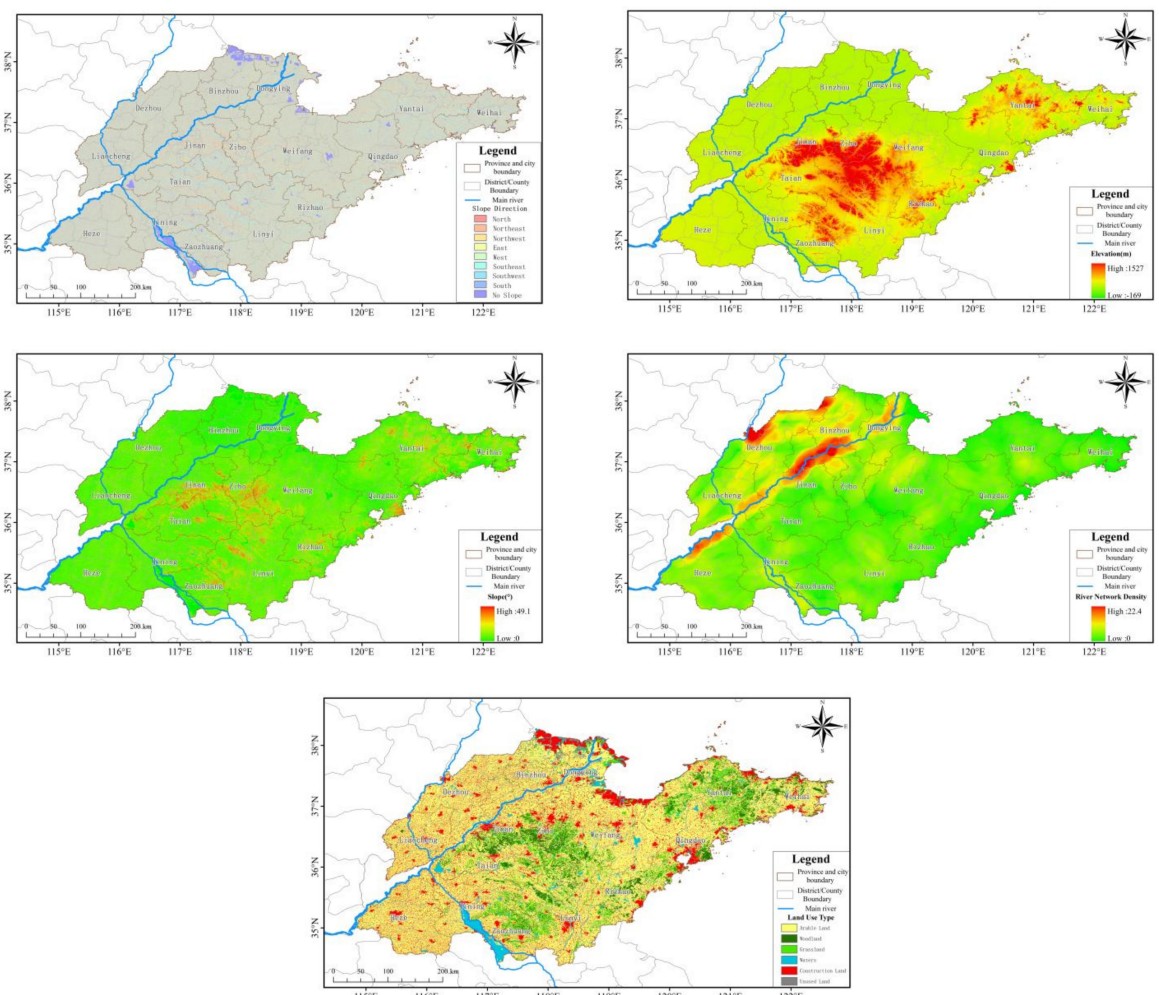

**Figure 5.** Spatial distribution map of slope aspect, altitude, slope, river network density, and land-use type in Shandong province.

**Table 7.** Grading and score of slope direction.

| Aspect | South | South West | South East | No Slope | East | West | Northeast | North West | North |
|--------|-------|------------|------------|----------|------|------|-----------|------------|-------|
| Score | 8 | 7 | 6 | 5 | 4 | 3 | 2 | 1 | 0 |

**Table 8.** Grading and score of the slope.

| Slope | 50–40° | 40–30° | 30–20° | 20–10° | 10–0° |
|---|---|---|---|---|---|
| Score | 5 | 4 | 3 | 2 | 1 |

**Table 9.** Land-use type scores.

| Land Use Type | Arable land | Woodland | Grassland | Waters | Construction Land | Unused Land |
|---|---|---|---|---|---|---|
| Score | 3 | 2 | 4 | 1 | 5 | 6 |

Adding the value of each factor according to its weight, the spatial distribution of the environmental sensitivity risk in Shandong province is obtained, as shown in Figure 6. There is little spatial difference in the environmental sensitivity from dry-hot wind disasters in Shandong province, yet the spatial distribution is uneven and shows a high degree of fragmentation.

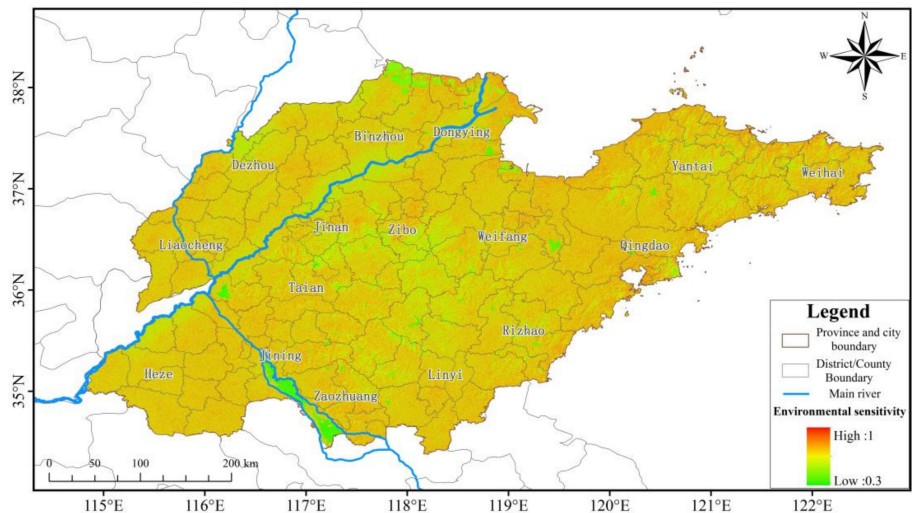

**Figure 6.** Spatial distribution of environmental sensitivity to dry-hot wind pregnancy in Shandong province.

### 3.1.3. Zoning of Dry-Hot Wind Risks

Meteorological factor risk and disaster environmental sensitivity are added according to their weight, and then classified to obtain the spatial distribution map of dry-hot wind disasters in Shandong province, as shown in Figure 7. Note that dry-hot wind disasters in Shandong province have clear spatial characteristics. The dry-hot winds at all levels in the north central area of Shandong province are all high-value areas, and the slope in the north central area is dominated by the southern slope with low altitude, small terrain slope, low river network density, and a large proportion of construction land. The sensitivity of the north central is higher. To sum up, the risk of dry-hot wind is higher in the north central area of Shandong province, with a distribution area of 32,587.6 km², accounting for 20.6% of the total land area. These areas include Dongying, Weifang, Zibo, east of Jinan, and south of Binzhou. The medium-risk regions are distributed in the peripheral area of the high-risk regions, including northwest of Binzhou, west of Jinan, Tai'an, Linyi, Rizhao, and east of Weifang. The medium-risk area comprises 6649.7 km², accounting for 42.1% of the total land area. The low-risk regions are in the west, southwest, and east, including Weihai, Yantai, and Qingdao on the Jiaodong Peninsula, and Heze, Jining, and Zaozhuang in the southwest. The low-risk area comprises 28,820.6 km², accounting for 37.3% of the total land area.

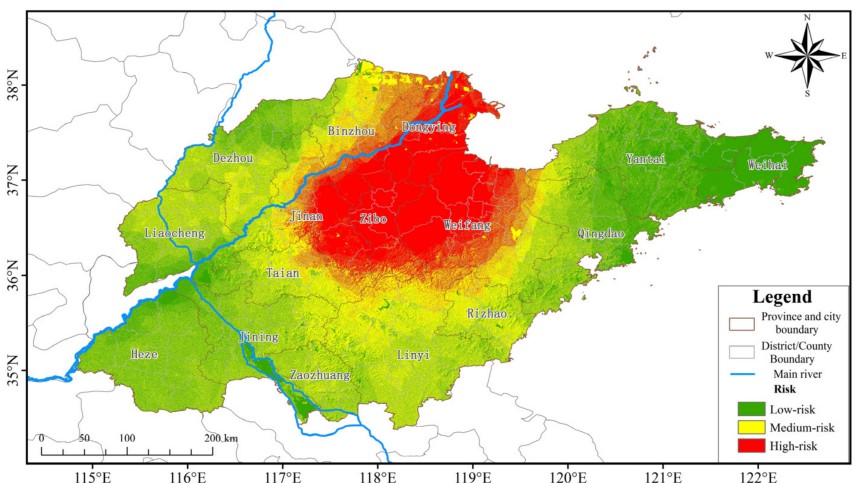

**Figure 7.** Spatial distribution of dry-hot wind hazards in Shandong province.

### 3.2. Spatial Distribution of Exposure of Disaster-Bearing Bodies

Exposure includes agricultural exposure and economic exposure. Crops are directly impacted by dry-hot wind disasters, and planting area directly reflects the degree of exposure. Thus, the sown area of crops is selected as the index of agricultural exposure. When a dry-hot wind disaster occurs, the higher the total GDP, the greater the total population, and the larger the administrative area, the stronger exposure will be from the disaster. Therefore, total GDP, total population, and administrative area are selected as economic exposure indices and added according to their weights in Figure 3 to obtain a spatial distribution map of economic exposure. Then, agricultural exposure and economic exposure are added with a weight of 0.7 and 0.3, respectively, and then classified to obtain the spatial zoning results of dry-hot wind exposure in Shandong province (Figure 8). Note that exposure to dry-hot wind disasters shows a clear spatial distribution pattern: generally high in the south and low in the north. High-exposure areas include Heze, Jining, Linyi, and Weifang, and medium-exposure areas are mainly in Dezhou, Liaocheng, Tai'an, Jinan, Yantai, and Qingdao. Low-exposure areas include Binzhou, Dongying, Zibo, Zaozhuang, Rizhao, and Weihai. The areas of high, medium, and low exposure are 56,581.8 km$^2$, 62,815.1 km$^2$, and 38,503.2 km$^2$, accounting for 35.8%, 39.8%, and 24.4% of the total land area, respectively.

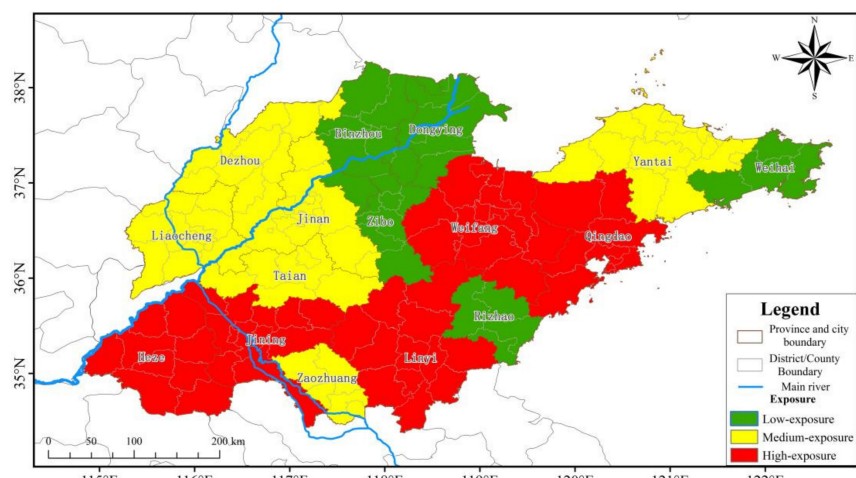

**Figure 8.** Spatial distribution of exposure of dry-hot wind-bearing bodies in Shandong province.

### 3.3. Spatial Distribution of Vulnerability of Disaster-Bearing Bodies

Vulnerability includes agricultural vulnerability and economic vulnerability. Dry-hot wind disasters hinder the grain filling of wheat and forces it to ripen, affecting its

maturation, and the thousand-grain weight is significantly reduced, resulting in a serious reduction in wheat production [38]. Therefore, the larger the wheat planting area, the greater the impact of dry-hot wind disasters. The present study uses the wheat planting area as the index of agricultural vulnerability. As for economic vulnerability, crop area proportion and population density are selected as indices of economic vulnerability. Crop area proportion refers to the ratio of crop planting area to administrative area. The larger the crop area proportion, the higher the vulnerability. The higher the population density, the higher the vulnerability. Crop area proportion and population density are spatially superimposed, as shown in Figure 3, to obtain the distribution map of economic vulnerability (not shown). Agricultural vulnerability and economic vulnerability are then added with a weight of 0.7 and 0.3, respectively, and classified to obtain the spatial zoning results of the vulnerability of disaster-bearing bodies (Figure 9). Note that vulnerability is high in the west and low in the east. The high-vulnerability areas include Heze, Liaocheng, and Dezhou in the west. The low vulnerability areas are Dongying, Zibo, Rizhao, Yantai, and Weihai. The remaining cities show medium vulnerability. The areas of high, medium, and low vulnerability are 31,503.3 km$^2$, 88,001.4 km$^2$, and 38,395.3 km$^2$, accounting for 20.0%, 55.7%, and 24.3% of the total land area, respectively.

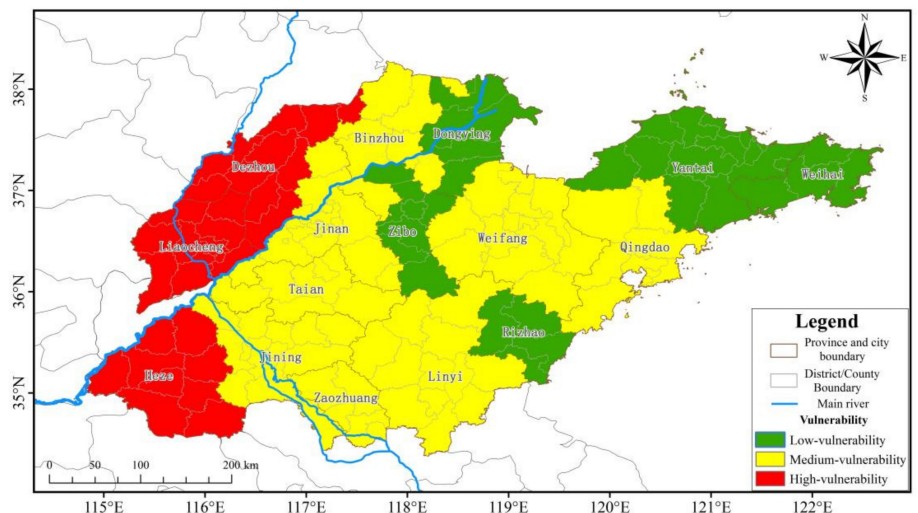

**Figure 9.** Spatial distribution of vulnerability of dry-hot wind-bearing bodies in Shandong province.

### 3.4. Spatial Distribution of Disaster Prevention and Mitigation Capability

Disaster prevention and mitigation capability refer to management measures and countermeasures used to prevent and reduce meteorological disasters. The higher the economic level of a place, the higher the disaster prevention and mitigation capability [39]. The higher the per capita GDP, per capita income, and level of education, the stronger the ability to defend against and respond to dry-hot wind disasters. Per capita GDP, per capita income, and education level are selected as indices of disaster prevention and mitigation capability. In accordance with the weights shown in Figure 3, the indices are added and then classified to obtain the spatial zoning results of the disaster prevention and mitigation capability in Shandong province (Figure 10). Note that the disaster prevention and mitigation capability of dry-hot wind disasters is high in the east and low in the west. Areas with high disaster prevention and mitigation capability are located mainly in Yantai, Weihai, Jinan, Qingdao, Jinan, and Dongying. Regions with low disaster prevention and mitigation capability include Dezhou, Liaocheng, Heze, Zaozhuang, and Linyi. The remaining areas are medium-risk regions. Areas with high, medium, and low disaster prevention and mitigation capability comprise 48,613.6 km$^2$, 56,086.5 km$^2$, and 53,199.8 km$^2$, accounting for 30.8%, 35.5%, and 33.7% of the total area, respectively.

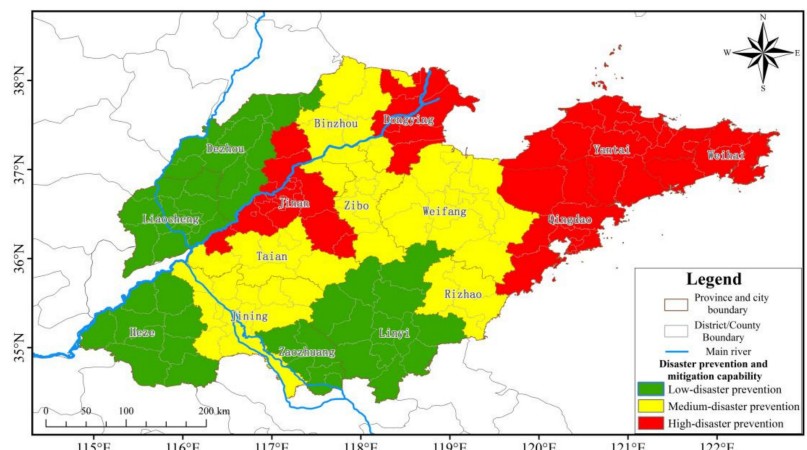

**Figure 10.** Spatial distribution of disaster prevention-mitigation capacity of dry and hot winds in Shandong province.

*3.5. Spatial Distribution of Comprehensive Risk of Dry-Hot Wind Disasters*

The four factors of the zoning results of risk, exposure of disaster-bearing bodies, vulnerability of disaster-bearing bodies, and disaster prevention and mitigation capability are spatially superimposed according to the weights shown in Figure 3. This obtains the zoning results of the comprehensive risk of dry-hot wind disaster in Shandong province, as shown in Figure 11. Note that the comprehensive risk of dry-hot wind disaster differs substantially in different areas. The medium- and high-risk regions are located mainly in the west and central areas, with low-risk regions in the east. The overall spatial distribution shows a strong degree of fragmentation. Table 10 shows the areas of the high-, medium-, and low-risk regions in each city. In Jining, Weifang, Heze, and Linyi, the area of high risk is largest. In Dongying, Qingdao, Rizhao, Weihai, Yantai, and Zaozhuang, there are no high-risk regions. Jining, Jinan, Tai'an, Binzhou, Linyi, and Dongying are medium-risk regions. The area of medium risk in Heze and Weihai is zero. In Yantai, Qingdao, and Weihai, the area of low risk is largest. In summary, there are no low-risk regions in Jining, Dezhou, Heze, Liaocheng, Linyi, and Weifang, only medium- and high-risk areas. The area of high-, medium-, and low-risk regions for dry-hot wind disasters is 64,076.7 km², 58,474.3 km², and 35,349.0 km², accounting for 40.6%, 37.0%, and 22.4% of the total land area, respectively.

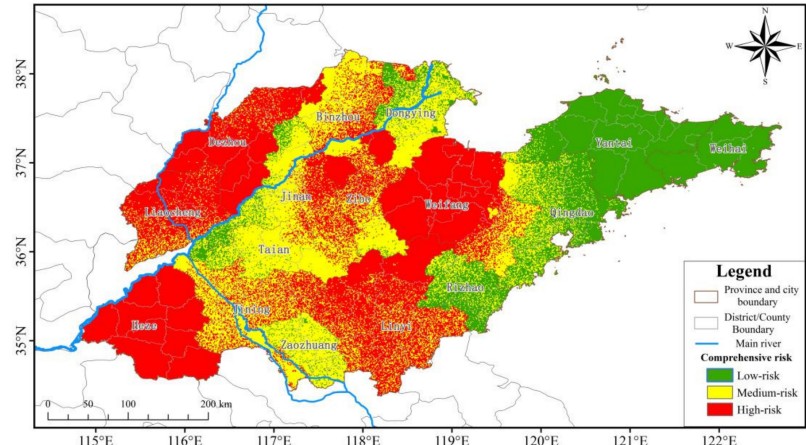

**Figure 11.** Spatial distribution of integrated riskiness of dry-hot wind in Shandong province.

**Table 10.** Area of high-, medium-, and low-risk areas in Shandong province by city.

| | High-Risk Area | | Medium-Risk Area | | Low-Risk Area | |
|---|---|---|---|---|---|---|
| | Area (km$^2$) | Ratio (%) | Area (km$^2$) | Ratio (%) | Area (km$^2$) | Ratio (%) |
| Binzhou City | 4028.6 | 2.6 | 5582.1 | 3.6 | 1 | 0 |
| Dezhou City | 9594.5 | 6.1 | 1019.1 | 0.7 | 0 | 0 |
| Dongying City | 0 | 0 | 5104.5 | 3.3 | 2064.5 | 1.3 |
| Heze City | 12,083.8 | 7.7 | 0 | 0 | 0 | 0 |
| Jinan City | 1955 | 1.2 | 7069.6 | 4.5 | 1443.9 | 0.9 |
| Jining City | 297,514 | 1.9 | 8184 | 5.2 | 0 | 0 |
| Liaocheng City | 6488.6 | 4.1 | 2263.2 | 1.4 | 0 | 0 |
| Linyi City | 11,891.2 | 7.6 | 5329.6 | 3.4 | 0 | 0 |
| Qingdao City | 0 | 0 | 3915.1 | 2.5 | 7276.5 | 4.6 |
| Rizhao City | 0 | 0 | 1668 | 1.1 | 3679.8 | 2.3 |
| Taian City | 375.2 | 0.2 | 6609.2 | 4.2 | 875.1 | 0.6 |
| Weihai City | 0 | 0 | 0 | 0 | 5687.5 | 3.6 |
| Weifang City | 13,089.7 | 8.4 | 3027.8 | 1.9 | 0 | 0 |
| Yantai City | 0 | 0 | 323.5 | 0.2 | 13,652.5 | 8.7 |
| Zaozhuang City | 0 | 0 | 3958.7 | 2.5 | 581 | 0.4 |
| Zibo City | 1595 | 1 | 4420 | 2.8 | 87.2 | 0.1 |

## 4. Discussion

(1) This study evaluates and classifies the risk of dry-hot wind disasters in Shandong province from the perspectives of risk, exposure, vulnerability, disaster prevention and mitigation capability, and comprehensive risk. Similar studies are rare. Thus, the results of this study provide a framework for related research in this area. The comprehensive risk of dry-hot wind disasters obtained here is compared with existing studies. Li et al. [40] use daily maximum temperature, relative humidity at 14:00, and wind speed at 14:00 from 1961 to 2017, as well as winter wheat growth period data to analyze the spatiotemporal characteristics of the disaster-causing factors of dry-hot wind disasters in the Huanghai and Huaihai areas. Their results show that dry-hot wind disasters occur frequently in northern and western Shandong. In comparison, the present study shows that the comprehensive high-risk area of dry-hot wind disasters is located mainly in Binzhou, Zibo, and Weifang, which is north of Shandong province, Dezhou and Liaocheng in the west, and Heze, Jining, and Linyi in the south. Note that these results are consistent with the previous study.

(2) In the present study, based on observation data from meteorological stations, the dry-hot wind index R of each station is calculated. Then, the Kriging interpolation method in ARC-GIS is used to obtain the spatial distribution map. In addition to the Kriging interpolation method, there are also other spatial interpolation methods, such as the inverse distance weight interpolation method, spline function method, and trend surface analysis. The results of different interpolation methods are shown in Figure 12. The spatial distribution of the dry-hot wind index obtained by different interpolation methods differs. The results of the Kriging interpolation, inverse distance weight interpolation, and spline function methods are similar, whereas the results of trend surface analysis are quite different. According to the results from the Kriging interpolation, inverse distance weight interpolation, and spline function methods, the areas with a high dry-hot wind index are located in the north-central area of Shandong province. Existing studies show that the occurrence of hot-dry wind disaster is frequent in the central part of Shandong province, and less frequent in the west [18]. Huimin County (Binzhou City) and Yangjiaogou Town (Weifang City) are two high-incidence areas of dry-hot wind disasters. Dezhou, Liaocheng, Jining, and Heze are low-risk areas. Yanzhou (Jining City) has a significantly higher occurrence frequency than Heze. In addition, the frequency of dry-hot wind disasters in Tai'an City increases from southwest to northeast [28]. Lin et al. [41] analyzed the weather data and hazard symptom information during the later stage of wheat growth, weather data for dry-hot wind days in past years, and field test data. These authors found that dry-hot wind

disasters occurred in Dezhou, Heze, Weifang, and Jining. The results obtained by Kriging interpolation are consistent with the results of previous studies.

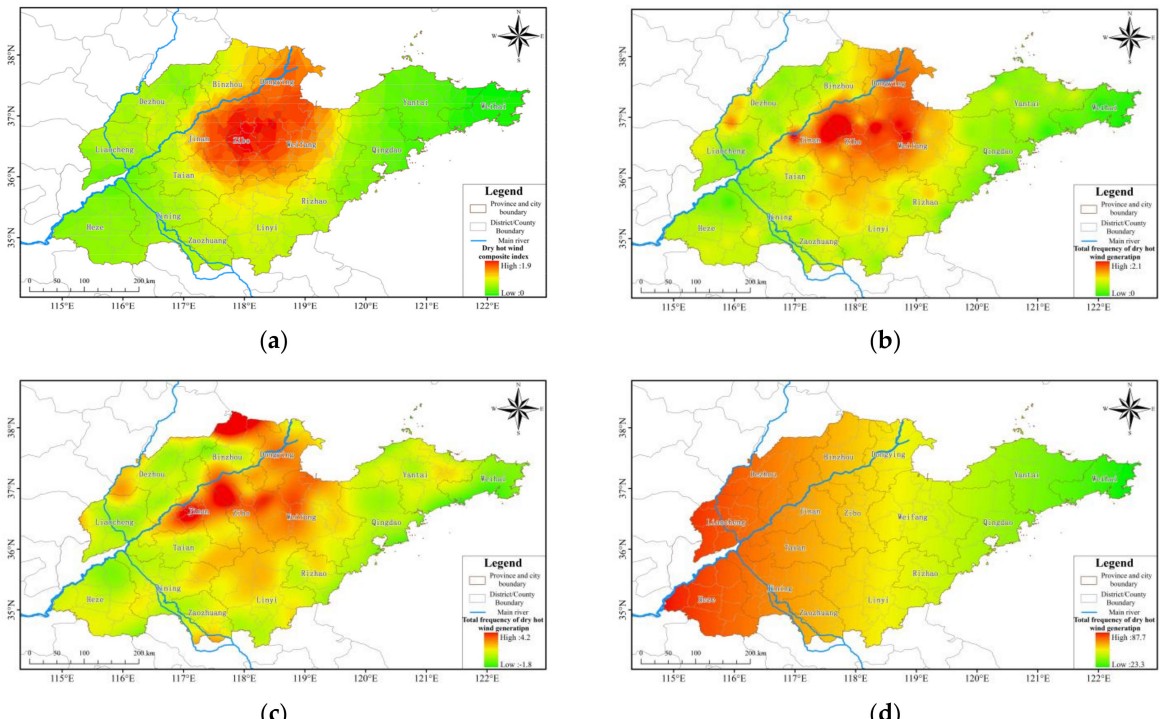

**Figure 12.** Spatial distribution of dry-hot wind index: (**a**) Kriging interpolation method; (**b**) inverse distance weight interpolation method; (**c**) spline function method; (**d**) trend surface analysis method.

(3) To perform the zoning of exposure, vulnerability, and disaster prevention and mitigation capability, only social and economic indicators are used in this study. However, the indicators in different counties are not standardized. For example, some indicators may not be present for certain counties. For the purpose of standardization, adjustments or replacements are made based on specific conditions, with varying effects on the zoning results. Moreover, some indicators are limited to the city level, and there is no county-level data, which affects the spatial resolution of the zoning. In addition, some indicators are not included in the statistical yearbooks; these are replaced with similar indicators. For instance, education level is supposed to be the percentage of graduation at each level, yet in the statistical yearbook, there is no relevant information. Thus, the number of school students is used to calculate education level. Apparently, such treatment affects the zoning results.

(4) The zoning results for dry-hot wind disasters include not only comprehensive risk, but also zoning for risk, exposure, vulnerability, and disaster prevention and mitigation capabilities. Therefore, in practical application, the results can be analyzed from various perspectives. For example, from the perspective of comprehensive risk zoning of dry-hot wind disasters (Figure 11), the comprehensive risk is high in Dezhou, Liaocheng, and Heze. However, when the zoning results of risk, exposure, vulnerability, and disaster prevention and mitigation capability are considered separately, we find that for areas with high comprehensive risk, such as Weifang, Jinan, Zibo, and Binzhou, disaster prevention awareness should be enhanced, and scientific disaster prevention and relief plans should be formulated. For areas with high exposure, such as Heze, Weifang, Linyi, and Jining, investment in disaster relief facilities should be increased. For Dezhou, Liaocheng, and Heze, which have high vulnerability and weak disaster prevention and mitigation capability, the focus should be on strengthening economic development and increasing investment in education, so as to reduce losses caused by dry-hot disaster. In conclusion, the risk assessment and zoning of dry-hot wind disasters in Shandong province can not only improve our understanding

of such disasters, but they can also provide a framework for government to formulate disaster prevention and relief policies.

(5) After calculating the risk index, exposure index, vulnerability index, disaster prevention and mitigation capabilities index, and comprehensive risk index, the risk space should be divided according to the size of the index. There are many methods to divide the index, such as the equal interval method, defined interval method, natural breakpoint classification method, standard deviation method, and so on. In the classification, we take the risk as an example and select four methods for comparison (Figure 13). It can be seen that the spatial differences of the zoning results obtained by the equal interval method (a), defined interval method (b) and standard deviation method (d) are not obvious, which means a smaller high-risk area in Figure 13a, a too-large risk area in Figure 13b, and the uncleared risk boundary in Figure 13d. The natural breakpoint classification method, compared with other methods, is the method with zoning results where spatial distribution of each grade is clear and is more consistent with the reality. Therefore, the natural breakpoint method is selected. The results of various division methods are as follows, and the classification scope of each index is added to the corresponding research content of the article.

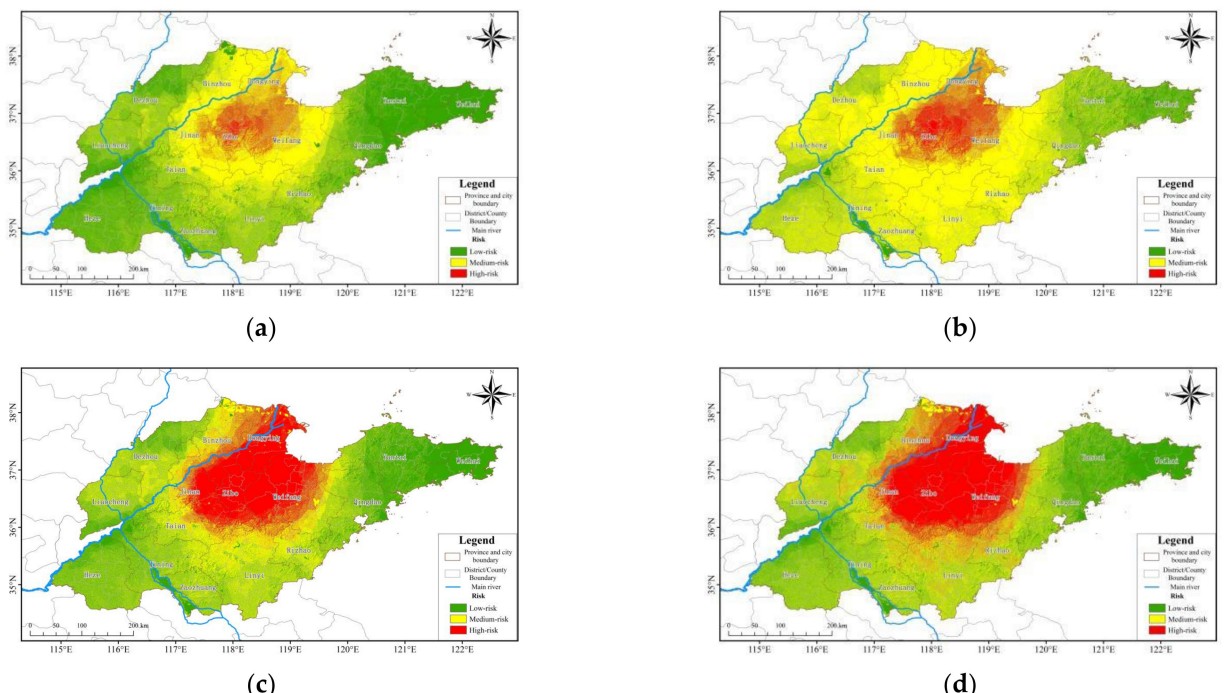

**Figure 13.** Spatial distribution of dry-hot wind hazards in Shandong province: (**a**) Equal Interval; (**b**) defined Interval; (**c**)natural breaks; (**d**) standard deviation.

(6) The dry-hot wind mainly reduced the wheat yield by reducing the 1000-grain weight of wheat. Therefore, this study calculated the average wheat yield per unit area of each administrative city (county) based on the statistical yearbook of each city in Shandong province in the past five years, and its spatial distribution is as follows.

Compared to the results of the dry-hot wind comprehensive risk zoning (Figure 11) and the yield per unit area of wheat, it can be seen that the spatial distribution of the two figures is basically similar (Figure 14). In areas with a high comprehensive risk of dry-hot wind, such as Dezhou, Liaocheng, Heze in the west, the yields per unit area of wheat are also the lowest; Dongying, Jinan, Tai'an, Rizhao, Qingdao, Weihai and other counties (cities) with low comprehensive risk of dry-hot wind, and the yields per unit area of wheat are also higher. However, there are also individual areas that do not match. For example, in the southwest of Dongying, the comprehensive risk of dry-hot wind is the highest, and the

average yield per unit area of wheat is also high, which may be caused by several factors, such as artificial irrigation.

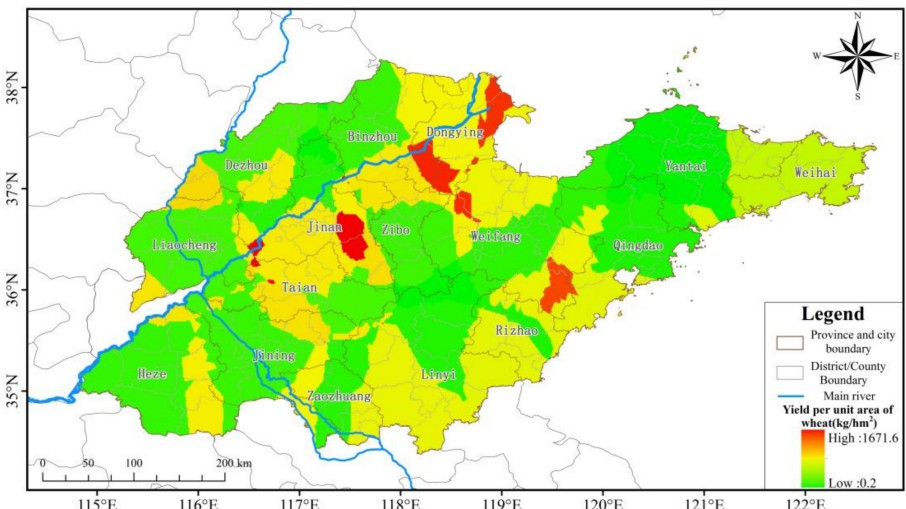

**Figure 14.** Spatial distribution of wheat yield per unit area in Shandong province.

## 5. Conclusions

The dry-hot wind, at all levels in Shandong province, mainly occurs in the central area of Shandong province, and the number of dry-hot wind in a year is at most 1.9 days. Severe dry-hot wind mainly occurs in the northern part of the central region, specifically in Zibo City, Weifang City, and Dongying City, where it occurs, at most, for 0.4 d. Considering the topographic factors, the high-value area of dry-hot wind risk index is located in the north-central area, with an area of 32,587.6 km$^2$, accounting for 20.6% of the province's area, and other areas are gradually decreasing. The high-value area of the exposure index is located in the southwest of Shandong province, covering an area of 56,581.5 km$^2$, accounting for 35.8% of the province's area. The high-risk areas in Zibo and Dongying are all low-exposure, and only Weifang is high-exposure. The high-value area of the dry-hot wind disaster-affected body vulnerability index in Shandong province is located in the west of Shandong province, with an area of 31,503.3 km$^2$, accounting for 20.0% of the province's area, while the high-value area of dry-hot wind risk has relatively low vulnerability. The high-value areas of the dry-hot wind disaster prevention and mitigation capabilities index in Shandong province are located in the eastern and central parts of Shandong province, with an area of 48,613.6 km$^2$, accounting for 30.8% of the province's area, and the lowest are in the southern and western regions. The disaster prevention and mitigation capabilities of the high-risk dry-hot wind area belongs to the medium area, among which Dongying City has the strongest disaster prevention and mitigation capabilities for dry-hot wind. To sum up, the high-value areas of dry-hot wind comprehensive risk in Shandong province are located in the western and central parts of Shandong province, with an area of 64,076.7 km$^2$, accounting for 40.6% of the province's area. Due to its strong disaster prevention and mitigation capabilities, Dongying City has become a low-value area in terms of comprehensive risk. Both Zibo City and Weifang City belong to areas with high comprehensive risk of dry-hot wind.

The study results showed that the comprehensive risk zoning results of dry-hot wind proposed in this paper were basically consistent with the spatial distribution of wheat yield per unit area in Shandong province, indicating that the results of dry-hot wind zoning in this study were of high accuracy. The results of this paper have important guiding significance for the formulation of disaster prevention and reduction planning of dry-hot wind in Shandong province. It is suggested to strengthen the construction of dry-hot wind early warning systems in the central and western regions, including improving the accuracy of dry-hot wind prediction, cultivating wheat varieties resistant to high

temperature and low humidity in the central and western regions, especially strengthening the water conservancy construction in the central and western regions, and improving the disaster prevention and reduction capacity of dry-hot wind resistance, so as to reduce the losses caused by strong exposure and vulnerability. In particular, Weifang City, Dezhou City, and Heze City should formulate different disaster prevention and reduction plans according to different mechanisms leading to high comprehensive risk of dry-hot wind. For example, Weifang is in a high-risk area.

**Author Contributions:** Writing—original draft preparation, N.W.; resources, X.X.; writing—review and editing, L.Z.; data curation, Y.C.; investigation, M.J.; software, Y.W. and Y.Z.; validation, X.G. and Y.Y.; visualization, E.Z. All authors have read and agreed to the published version of the manuscript.

**Funding:** This research was funded by the National Natural Science Foundation of China (Grant No. 41771067) and the Key projects of Natural Science Foundation of Heilongjiang province of China (No. ZD2020D002).

**Institutional Review Board Statement:** Not applicable.

**Informed Consent Statement:** Not applicable.

**Data Availability Statement:** Not applicable.

**Conflicts of Interest:** The authors declare no conflict of interest.

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
