# Peer review of "Spatial Zoning of Dry-Hot Wind Disasters in Shandong Province"

_sustainability, doi:10.3390/su14073904_

Round 1

Reviewer 1 Report

The development of methods and indices useful to evaluate the vulnerability of a territory to weather extreme events is really interesting, also due to the relevance of the climate change issue. Although the authors expressed aim, method and results clearly, following there are some aspects that need to be improved:

section 1: the scientific framework is missing. This section introduces the main topic, but without providing readers with useful references to previous studies, reports, and so on that have already dealt with the same topic. 

figure 2: how were the weights of risk, exposure and vulnerability calculated? The authors used ahp, but going through the numerous subsections of 2.3.1 I missed the way these 3 weights were obtained. At the same way, please clarify the coefficients used in formula n.9.

section 5: the authors could refer their results to planning system of the study area. Their results are useful in this perspective, so in this last section they could better underline this aspect.

Author Response

My modifications are in the file below.

Reviewer 2 Report

This manuscript describes the risk of dry-hot wind on Shandong Province. The structure of the paper is well-organized and the topic fits the scope of the journal. The reviewer would like to suggest that this article can be accepted after some questions are clarified. Some comments are listed below.

  1. Line 30: The physical causing of dry-hot wind should be mentioned.
  2. Line 170: The reference and introduction of the method of Analytic Hierarchy Process (AHP) is required.
  3. Line 227: The application of the Kriging interpolation is not clear. When does it be used? Does the Kriging interpolation applied respectively to all the influenced factors in advance or just for the risk map? The statement should be added in the article.
  4. Table 4 and table 5: The reviewer is confused about the grading and the score of slope and slope direction. How does the value be conducted? Is there any criterion between scales?
  5. Figure 5: The authors can consider to have some statement for the attribution of the spatial distribution of dry-hot wind hazard.

Author Response

My modifications are in the file below.

Reviewer 3 Report

In this study, the spatial zoning of dry-hot wind disaster was examined. The theoretical model and statistical analysis methods used in the study were properly explained. I understand the process of examination, but I do not understand the validity of the process. So, I request the authors to add the following explanation.

In lines 220 and 222, while it is appropriate in some sense, is the index always linear in its relation to disaster risk? The relationship between indices and risk needs to be modeled based on observed disaster data. Please reconsider the modeling approach for the relationship between indices and risk.

In line 227, please explain the validity of using Kriging interpolation as a method of spatial reclassification in this study.

In line 234, please provide additional information about the respondents of the AHP method. Please explain the reasons and validity of the selection of experts.

In line 246, please explain the reasons for the selection of D1, Dm, Ds and please explain the basis for the weights of 0.2, 0.3, and 0.5. How did you deal with days when D1, Dm, and Ds occur simultaneously? And, please show the relationship between the predicted value R using the equation (9) and the actual observed value, and provide the prediction accuracy.

In lines 264-270, please indicate the basis for air temperature, humidity, wind speed, precipitation that distinguish ‘mild’, ‘moderate’, ‘severe’, and ‘composite’, based on actual disaster data.

In lines 284-287, please explain the reasons why the scores are linearly determined for each category. The relationship between indices and risk needs to be modeled based on observed disaster data. Please reconsider the modeling approach for the relationship between indices and risk.

Please explain the reasons based on actual disaster data for the distinction between low-risk, medium-risk, high-risk in Figure 5, low-exposure, medium-exposure, high-exposure in Figure 6, low-vulnerability, medium-vulnerability, high-vulnerability in Figure 7, low-disaster prevention, medium-disaster prevention, high-disaster prevention in Figure 8, low-risk, medium-risk, high-risk in Figure 9, respectively.

Author Response

My modifications are in the file below.

Reviewer 4 Report

The research topic is interesting, but the aims of this study and validation should be improved as below:

1) The aims of this study should be addressed at the last paragraph of Introduction;

2) Validation should be mentioned in the result section.

More specifically:

1) writing is quite good;

2) Most figures are fine except for an unclear Figure 3;

3) All figures should be added with a scale bar;

4) All tables are fine;

5) Some latest references should be updated. 

Round 2

Reviewer 3 Report

As shown below, the reviewers did not respond appropriately to my previous remarks, therefore I am unable to change the results of my previous evaluation.

1, 6. The authors agreed with my opinion, but did not provide the relationship between indices and risk that I requested.

2. The authors added wording in response to my point, but did not provide the explanation I requested. Variables such as air temperature may be proportional to the square or one-half power of the distance from an influencing factor such as a heat source. I asked the authors for physical insight rather than a technical explanation of the kriging method.

3. The process by which the weights were determined to be 0.3 and 0.7 based on expert opinion was not shown.

4. The reasons for the weights of about 0.2, 0.3, and 0.5 for light, medium, and heavy dry hot wind were not explained.

5. The distinction between 'mild', 'moderate', 'severe', and 'composite' was not explained.

7. The extent of each classification and its reason were not explained.

Reviewer 4 Report

The revised version has been improved well to meet the requirement of the journal so that it is agreed to be accepted.